EMBO
Molecular Medicine

# sTREM2 cerebrospinal fluid levels are a potential biomarker for microglia activity in early-stage Alzheimer's disease and associate with neuronal injury markers

Marc Suárez-Calvet[1,2], Gernot Kleinberger[1,3], Miguel Ángel Araque Caballero[4], Matthias Brendel[5], Axel Rominger[3,5], Daniel Alcolea[6,7], Juan Fortea[6,7], Alberto Lleó[6,7], Rafael Blesa[6,7], Juan Domingo Gispert[8,9], Raquel Sánchez-Valle[10,11], Anna Antonell[10,11], Lorena Rami[10,11], José L Molinuevo[8,9,10,11], Frederic Brosseron[12], Andreas Traschütz[13], Michael T Heneka[12,13], Hanne Struyfs[14,15], Sebastiaan Engelborghs[14,15], Kristel Sleegers[16,17], Christine Van Broeckhoven[16,17], Henrik Zetterberg[18,19], Bengt Nellgård[20], Kaj Blennow[18], Alexander Crispin[21], Michael Ewers[4,*,†] & Christian Haass[1,2,3,**,†]

## Abstract

TREM2 is an innate immune receptor expressed on the surface of microglia. Loss-of-function mutations of *TREM2* are associated with increased risk of Alzheimer's disease (AD). TREM2 is a type-1 protein with an ectodomain that is proteolytically cleaved and released into the extracellular space as a soluble variant (sTREM2), which can be measured in the cerebrospinal fluid (CSF). In this cross-sectional multicenter study, we investigated whether CSF levels of sTREM2 are changed during the clinical course of AD, and in cognitively normal individuals with suspected non-AD pathology (SNAP). CSF sTREM2 levels were higher in mild cognitive impairment due to AD than in all other AD groups and controls. SNAP individuals also had significantly increased CSF sTREM2 compared to controls. Moreover, increased CSF sTREM2 levels were associated with higher CSF total tau and phospho-tau$_{181P}$, which are markers of neuronal degeneration and tau pathology. Our data demonstrate that CSF sTREM2 levels are increased in the early symptomatic phase of AD, probably reflecting a corresponding change of the microglia activation status in response to neuronal degeneration.

**Keywords** Alzheimer's disease; biomarkers; microglia; neurodegeneration; TREM2
**Subject Categories** Biomarkers & Diagnostic Imaging; Neuroscience

See also: **SE Schindler & DM Holtzman** (May 2016)

1   BioMedical Center (BMC), Biochemistry, Ludwig-Maximilians-University Munich, Munich, Germany
2   German Center for Neurodegenerative Diseases (DZNE) Munich, Munich, Germany
3   Munich Cluster for Systems Neurology (SyNergy), Munich, Germany
4   Institute for Stroke and Dementia Research, Klinikum der Universität München, Ludwig-Maximilians-University Munich, Munich, Germany
5   Department of Nuclear Medicine, Klinikum der Universität München, Ludwig-Maximilians-University Munich, Munich, Germany
6   Department of Neurology, Institut d'Investigacions Biomèdiques, Hospital de la Santa Creu i Sant Pau, Universitat Autònoma de Barcelona, Barcelona, Spain
7   Center for Networked Biomedical Research for Neurodegenerative Diseases, CIBERNED, Madrid, Spain
8   Clinical and Neuroimaging Departments, Barcelona Beta Brain Research Center, Pasqual Maragall Foundation, Barcelona, Spain
9   Centro de Investigación Biomédica en Red de Bioingeniería, Biomateriales y Nanomedicina (CIBER-BBN), Barcelona, Spain
10  Alzheimer's Disease and Other Cognitive Disorders Unit, Neurology Service, ICN Hospital Clinic i Universitari, Barcelona, Spain
11  Institut d'Investigacions Biomèdiques August Pi i Sunyer (IDIBAPS), Barcelona, Spain
12  German Center for Neurodegenerative Diseases (DZNE), Bonn, Germany
13  Neurology Department, Universitätsklinikum Bonn, Bonn, Germany
14  Reference Center for Biological Markers of Dementia (BIODEM), Laboratory of Neurochemistry and Behavior, Institute Born-Bunge, University of Antwerp, Antwerp, Belgium
15  Department of Neurology and Memory Clinic, Hospital Network Antwerp (ZNA) Middelheim and Hoge Beuken, Antwerp, Belgium
16  Neurodegenerative Brain Diseases Group, Department of Molecular Genetics, VIB, Antwerp, Belgium
17  Laboratory of Neurogenetics, Institute Born-Bunge, University of Antwerp, Antwerp, Belgium
18  Clinical Neurochemistry Lab, Institute of Neuroscience and Physiology, The Sahlgrenska Academy at the University of Gothenburg, Mölndal, Sweden
19  Reta Lila Weston Laboratories and Department of Molecular Neuroscience, UCL Institute of Neurology, London, UK
20  Department of Anaesthesiology and Intensive Care, Institute of Clinical Sciences, Sahlgrenska Academy, Gothenburg University, Gothenburg, Sweden
21  Institute of Medical Informatics, Biometry, and Epidemiology, Munich, Germany
    *Corresponding author. Tel: +49 89 4400 46221; Fax: +49 89 4400 46113; E-mail: michael.ewers@med.uni-muenchen.de
    **Corresponding author. Tel: +49 89 4400 46549; Fax: +49 89 4400 46546; E-mail: christian.haass@mail03.med.uni-muenchen.de
    †These authors contributed equally to this study

## Introduction

Heterozygous missense mutations in the gene encoding the triggering receptor expressed on myeloid cells 2 (*TREM2*) have been recently described to significantly increase the risk of late onset Alzheimer's disease (AD) with an odds ratio similar to that of carrying an apolipoprotein E *(APOE)* ε4 allele (Guerreiro *et al*, 2013a; Jonsson *et al*, 2013). Heterozygous missense mutations in *TREM2* also increase the risk for other neurodegenerative diseases (Borroni *et al*, 2013; Rayaprolu *et al*, 2013; Cuyvers *et al*, 2014), and homozygous loss-of-function mutations in *TREM2* cause Nasu–Hakola disease (NHD) (Paloneva *et al*, 2002) and frontotemporal dementia (FTD)-like syndrome (Guerreiro *et al*, 2013b), which are both early-onset neurodegenerative diseases presenting as a frontal syndrome. Together, these findings indicate that TREM2 may be a common denominator in the pathogenesis of several different neurodegenerative diseases.

TREM2 is a type-1 transmembrane glycoprotein with an immunoglobulin-like extracellular domain, one transmembrane domain and a short cytosolic tail (Klesney-Tait *et al*, 2006). It belongs to the TREM family of innate immune receptors and is expressed in cells of the monocytic lineage (Bouchon *et al*, 2001; Schmid *et al*, 2002; Paloneva *et al*, 2003; Kiialainen *et al*, 2005). In the central nervous system (CNS), it is selectively expressed by microglia and involved in regulating phagocytosis and removal of apoptotic neurons as well as in the inhibition of microglia proinflammatory response (Takahashi *et al*, 2005; Klesney-Tait *et al*, 2006; Hsieh *et al*, 2009; Wang *et al*, 2015). We have shown that loss-of-function of TREM2 impairs the phagocytic activity of microglial cells and reduces clearance of amyloid β-peptide (Aβ; Kleinberger *et al*, 2014), suggesting that TREM2 may play an important role in the development of AD pathology and neurodegeneration during the course of the disease.

TREM2 undergoes proteolytic processing, releasing its ectodomain into the extracellular space as a soluble variant (sTREM2) via shedding by ADAM proteases (Wunderlich *et al*, 2013; Kleinberger *et al*, 2014), and can be detected in human plasma and cerebrospinal fluid (CSF) (Piccio *et al*, 2008; Kleinberger *et al*, 2014). Piccio *et al* found that CSF sTREM2 levels were increased in multiple sclerosis and other neurological inflammatory diseases (Piccio *et al*, 2008). We described that sTREM2 was almost undetectable in the CSF and plasma of a FTD-like patient carrying a homozygous *TREM2* p.T66M mutation. This mutation leads to misfolding of the full-length protein, which accumulates within the endoplasmic reticulum. Due to the lack of cell surface transport, shedding is dramatically reduced, which explains the absence of sTREM2 in patients with the homozygous *TREM2* p.T66M mutation (Kleinberger *et al*, 2014). In contrast, we observed a slight decrease in CSF sTREM2 levels in AD dementia patients compared to elderly cognitively normal subjects (Kleinberger *et al*, 2014). While our manuscript was under consideration, we learned that two groups independently found that CSF sTREM2 was increased in AD patients (Heslegrave *et al*, 2016; Piccio *et al*, 2016). Taken together, these results suggest that CSF sTREM2 levels are altered not only in subjects with *TREM2* mutations but also in sporadic cases of neurodegenerative diseases. In AD, amyloid plaques and neurofibrillary tangles, the major pathological hallmarks of the disease, develop decades before the onset of clinical symptoms (Morris *et al*, 1996; Braak & Braak, 1997;

Hulette *et al*, 1998; Price & Morris, 1999). Increased microglial activation and neuroinflammation frequently accompanies the early development of Aβ and tau pathology (Mosher & Wyss-Coray, 2014; Streit *et al*, 2014; Heneka *et al*, 2015; Tanzi, 2015). Since TREM2 is a key protein involved in the activation of microglia, the question arises whether TREM2 levels are pathologically altered in the early course of AD. If so, CSF sTREM2 would be an attractive biomarker candidate for tracking of the disease and as a potential outcome parameter for future clinical trials focusing on TREM2 and neuroinflammation. However, it is not known whether CSF sTREM2 levels change during the different stages of AD, a question that we addressed in the current study.

The main aim of this cross-sectional multicenter study was to determine whether the levels of CSF sTREM2 change across the continuum of AD. We tested CSF TREM2 in subjects with preclinical AD, mild cognitive impairment (MCI) due to AD (MCI-AD), and AD dementia and controls, defined by clinical and CSF biomarker criteria as recommended by the National Institute on Aging-Alzheimer's Association (NIA-AA) criteria (Albert *et al*, 2011; McKhann *et al*, 2011; Sperling *et al*, 2011). We also tested whether CSF sTREM2 levels are associated with the core AD CSF biomarkers $A\beta_{1-42}$, total tau (T-tau) and tau phosphorylated at threonine 181 (P-tau$_{181P}$) (Blennow *et al*, 2010). As a secondary aim, we investigated whether CSF sTREM2 levels are altered in cognitively normal subjects with suspected non-AD pathology (SNAP) and MCI subjects without CSF biomarker evidence of AD pathology (MCI-noAD). SNAP is a recently defined diagnostic category that comprises those individuals with abnormal neurodegeneration biomarkers (T-tau and P-tau$_{181P}$) but without evidence of underlying amyloidosis (Jack *et al*, 2012) and might thus represent neurodegenerative diseases different from AD.

## Results

### Study population

The current study included 150 controls and 63 preclinical AD, 111 MCI-AD as well as 200 AD dementia subjects (Table 1). The diagnostic criteria of each group were defined according to the NIA-AA criteria, which uses a combination of clinical diagnosis and the CSF biomarker profile including $A\beta_{1-42}$, T-tau, and P-tau$_{181P}$ (Albert *et al*, 2011; McKhann *et al*, 2011; Sperling *et al*, 2011). Decreased $A\beta_{1-42}$ was a requisite for preclinical AD, and the combination of decreased $A\beta_{1-42}$ and increased T-tau and/or P-tau$_{181P}$ for MCI-AD and AD dementia. The control group consisted of asymptomatic cognitively normal individuals with all three AD CSF core biomarkers within the normal range. The diagnostic criteria are described in more detail in the methods section. The cutoff values to define abnormal CSF values for each of the three AD CSF core biomarkers were defined for each center and are displayed in Appendix Table S1 (Antonell *et al*, 2011; Alcoea *et al*, 2014; Van der Mussele *et al*, 2014).

The demographic and CSF core biomarkers values of control, preclinical AD, MCI-AD, and AD dementia subjects are shown in Table 1. All patients in the AD continuum group were older and had a higher frequency of *APOE* ε4 carriers than the control group.

Age and *APOE* ε4 status did not differ between the three AD subcategories. As expected, groups differed with regard to their CSF biomarkers profiles. There were no differences in gender between groups.

### CSF sTREM2 is influenced by age

Age was positively correlated with CSF sTREM2 in the pooled group of subjects (Pearson $r = +0.391$, $P < 0.0001$). The correlation was still significant when tested within each diagnostic group, including the control group (Pearson $r = +0.177$, $P = 0.030$), preclinical AD (Pearson $r = +0.510$, $P < 0.0001$), MCI-AD (Pearson $r = +0.289$, $P = 0.002$), and AD dementia (Pearson $r = +0.310$, $P < 0.0001$) (Fig 1). Levels of CSF sTREM2 were not significantly affected by gender ($F_{1,521} = 0.1$, $P = 0.719$) nor by *APOE* ε4 status ($F_{1,377} = 0.4$, $P = 0.552$), controlled by age and center, when tested within the entire sample or each diagnostic group.

### Increase of CSF TREM2 levels in MCI due to AD

CSF sTREM2 differed between diagnostic groups, controlled for age, gender, and center ($F_{3,517} = 4.9$, $P = 0.002$). Levels of CSF sTREM2 were significantly higher in MCI-AD than in controls ($P = 0.002$) and AD dementia ($P = 0.013$) groups. There was a tendency for higher CSF sTREM2 levels in MCI-AD compared to preclinical AD ($P = 0.062$). No other group differences were found (Fig 2 and Table 2). In order to confirm the robustness of the results, we also calculated the means and 95% CI of CSF sTREM2 in each diagnostic group based on 1,000 bootstrap samples and the group comparison based on the overlap of the 95% CI confirmed the significant increase of CSF sTREM2 in MCI-AD compared to the control and AD dementia groups.

Given that the control group was significantly younger, we took additional measures to ensure that the group differences in CSF TREM2 between the control and the MCI-AD groups were not attributable to age differences. We repeated the regression analysis, this time restricted to those subjects who were 65 years or older. In this subgroup of older subjects, CSF sTREM2 remained significantly increased in MCI-AD compared to the control group ($P = 0.0005$), despite the fact that the ages between the two groups were not significantly different (Appendix Table S2).

From these findings, we conclude that CSF sTREM2 is increased in individuals with MCI-AD.

### Association between CSF TREM2 levels and core CSF biomarkers of AD

We studied the relationship between CSF sTREM2 and the core AD CSF biomarkers using linear mixed-effects models, controlled for age, gender, and center. In the whole sample of subjects including the controls and all the AD continuum groups, increased CSF sTREM2 was associated with higher levels of T-tau ($\beta = +0.336$, $P = 0.001$) (Fig 3A and B) and P-tau$_{181P}$ ($\beta = +0.370$, $P = 0.001$) (Fig 3C and D), and lower levels of Aβ$_{1-42}$ ($\beta = -0.098$, $P = 0.014$) (Fig 3E and F). Within each diagnostic group, the positive association between CSF sTREM2 and T-tau or P-tau$_{181P}$ was present (Fig 3A–D), except for the association between CSF sTREM2 and T-tau in the MCI-AD group ($\beta = +0.184$, $P = 0.098$). On the other hand, higher CSF sTREM2 levels showed a tendency to be associated with higher Aβ$_{1-42}$ in the control group ($\beta = +0.159$, $P = 0.060$), but with lower Aβ$_{1-42}$ in the MCI-AD group ($\beta = -0.291$, $P = 0.002$) (Fig 3E and F). We conclude that higher CSF sTREM2 correlates with higher levels of markers of neuronal injury and tau pathology (i.e. T-tau and P-tau$_{181P}$)

**Table 1.  Demographic and clinical characteristics of the control and AD continuum groups.**

| Variable | Control ($n = 150$) | AD continuum ($n = 374$) Preclinical AD ($n = 63$) | MCI-AD ($n = 111$) | AD dementia ($n = 200$) | *P*-value (group effect) |
|---|---|---|---|---|---|
| Females, % | 59 | 60 | 60 | 62 | 0.940 |
| *APOE* ε4 carriers, % | 21 | 58* | 52* | 62* | <0.0001 |
| Age, years | 62.4 (11) | 70.8 (11)* | 74.3 (9)* | 73.8 (10)* | <0.0001 |
| CSF biomarkers | | | | | |
| Aβ$_{1-42}$, pg/ml | 796 (159) | 414 (98)* | 426 (107)* | 408 (113)* | <0.0001 |
| T-tau, pg/ml | 218 (81) | 450 (428)$^{†}$ | 737 (410)$^{*,‡}$ | 920 (564)$^{*,§,¶}$ | <0.0001 |
| P-tau$_{181P}$, pg/ml | 43 (12) | 66 (39)* | 95 (32)$^{*,§}$ | 102 (44)$^{*,§}$ | <0.0001 |

Aβ, amyloid β-peptide; AD, Alzheimer's disease; APOE, apolipoprotein E; CSF, cerebrospinal fluid; MCI-AD, MCI due to AD; P-tau$_{181P}$, tau phosphorylated at threonine 181; T-tau, total tau.

Data are expressed as percent (%) or mean (SD), as appropriate. Probability values (*P*) denote differences between groups.

*APOE* genotype was available in 103 controls (69%), 39 preclinical AD (62%), 89 MCI-AD (80%), and 148 AD dementia (74%). Only Aβ$_{1-42}$ values measured by the INNOTEST ELISA are included; Aβ$_{1-42}$ values from Bonn group (measured with MSD platform) are excluded.

Chi-square statistics were used for the group comparisons of gender and *APOE* ε4 carrier. One-way ANOVA was used to compare age and CSF biomarkers between groups. The *P*-values indicated in the last column refer to the group effects in these tests. Significant group effects were followed by Bonferroni-corrected pair-wise *post hoc* tests.

*$P < 0.0001$ versus controls.
$^{†}P = 0.002$ versus controls.
$^{‡}P = 0.0001$ versus preclinical AD.
$^{§}P < 0.0001$ versus preclinical AD.
$^{¶}P = 0.002$ versus MCI-AD.

    

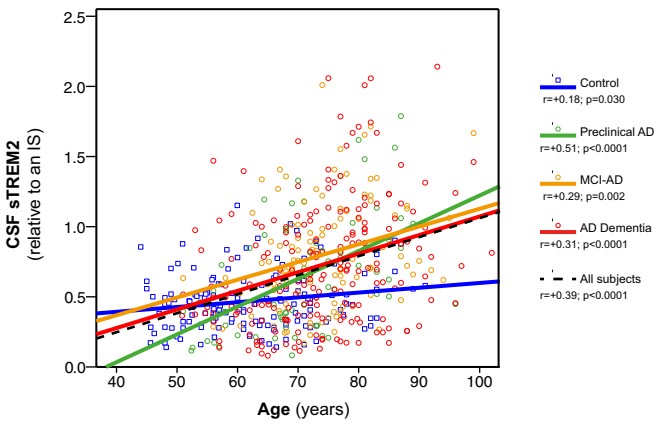

**Figure 1. CSF sTREM2 is associated with age.**
Scatter plot representing CSF sTREM2 as a function of age in the different diagnostic groups. Solid lines indicate the linear regression for each of the groups; the dashed line indicates the linear regression within the entire sample. *P*-values were assessed by Pearson product-moment correlations..

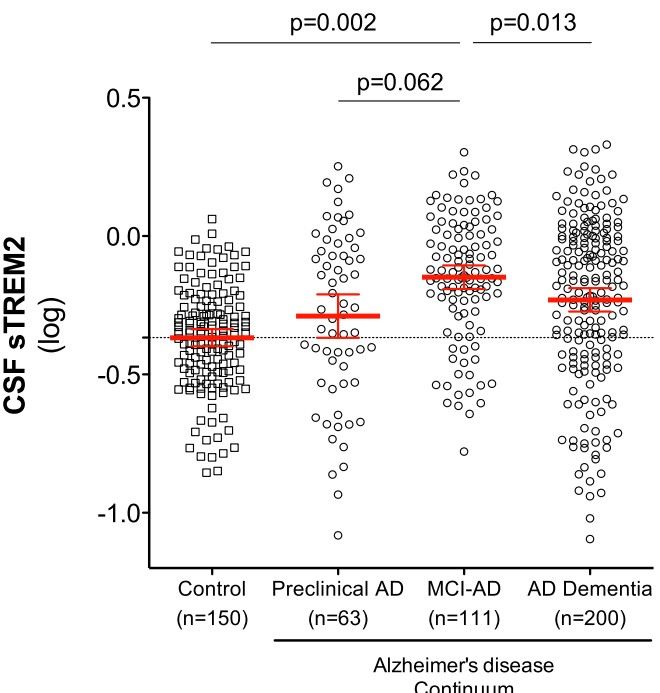

**Figure 2. CSF sTREM2 levels in the different diagnostic groups.**
Scatter plot showing levels of CSF sTREM2 (log-transformed) in the different diagnostic groups. Red bars represent the mean and the 95% CI. *P*-values were assessed by a linear mixed model adjusted by age and gender (fixed effects) and center (random effects).

suggesting an early response of TREM2 to first symptoms of neurodegeneration.

## CSF TREM2 levels in cognitively normal SNAPs and in MCI-noAD subjects

As a secondary aim, we studied CSF samples from cognitively normal individuals with SNAP ($n = 39$), that is, individuals with abnormal neurodegeneration biomarkers (T-tau and/or P-tau$_{181P}$) without the presence of significant amyloid pathology (as measured by abnormally decreased A$\beta_{1-42}$) (Jack *et al*, 2012). We compared them with the rest of cognitively normal individuals, namely controls and preclinical AD group. Interestingly, we found that CSF sTREM2 levels differed between groups ($F_{2,242} = 7.5$, $P = 0.0007$) and were particularly increased in the SNAP group compared to the control ($P = 0.0004$) and the

preclinical AD group ($P = 0.024$) (Fig 4A). These findings may indicate that an increase in CSF sTREM2 levels in response to neuronal injury (as measured by CSF tau levels) can occur without amyloidosis.

Next, we tested CSF TREM2 levels in subjects clinically diagnosed with MCI but showing no CSF biomarker profile of AD pathology (MCI-noAD), that is, biomarkers do not indicate a high likelihood that the MCI syndrome is due to AD (Albert *et al*, 2011).

**Table 2. Summary of the linear mixed model analysis with CSF sTREM2 as outcome variable and diagnostic group, gender, age, and center as predictor variables.**

| Diagnostic group | Unadjusted mean | 95% CI | Adjusted mean* | 95% CI | *n* |
|---|---|---|---|---|---|
| Control | −0.367 | −0.398, −0.337 | −0.294 | −0.387, −0.201 | 150 |
| Preclinical AD | −0.289 | −0.368, −0.211 | −0.273 | −0.371, −0.175 | 63 |
| MCI-AD | −0.149[†,‡§] | −0.191, −0.106 | −0.171[¶,**,††] | −0.265, −0.077 | 111 |
| AD dementia | −0.259[†] | −0.273, −0.188 | −0.261 | −0.353, −0.168 | 200 |

AD, Alzheimer's disease; CI, confidence interval; CSF, cerebrospinal fluid; MCI-AD, MCI due to AD.
CSF sTREM2 levels are expressed in their log-transformed values. They are shown as unadjusted means and 95% CI (*P*-values calculated by one-way ANOVA) and adjusted (*) for gender and age (fixed effects) and center (random effects) in a linear mixed model.
Adjustments based on age mean = 70.26. *Post hoc* comparisons (Bonferroni):
[†]$P < 0.0001$ versus control.
[‡]$P = 0.004$ versus preclinical AD.
[§]$P = 0.048$ versus AD dementia.
[¶]$P = 0.002$ versus control.
[**]$P = 0.062$ versus preclinical AD.
[††]$P = 0.013$ versus AD dementia.
Note that the increase of CSF sTREM2 in MCI-AD compared to the control and the AD dementia groups is still significant after adjusting by gender, age, and center.

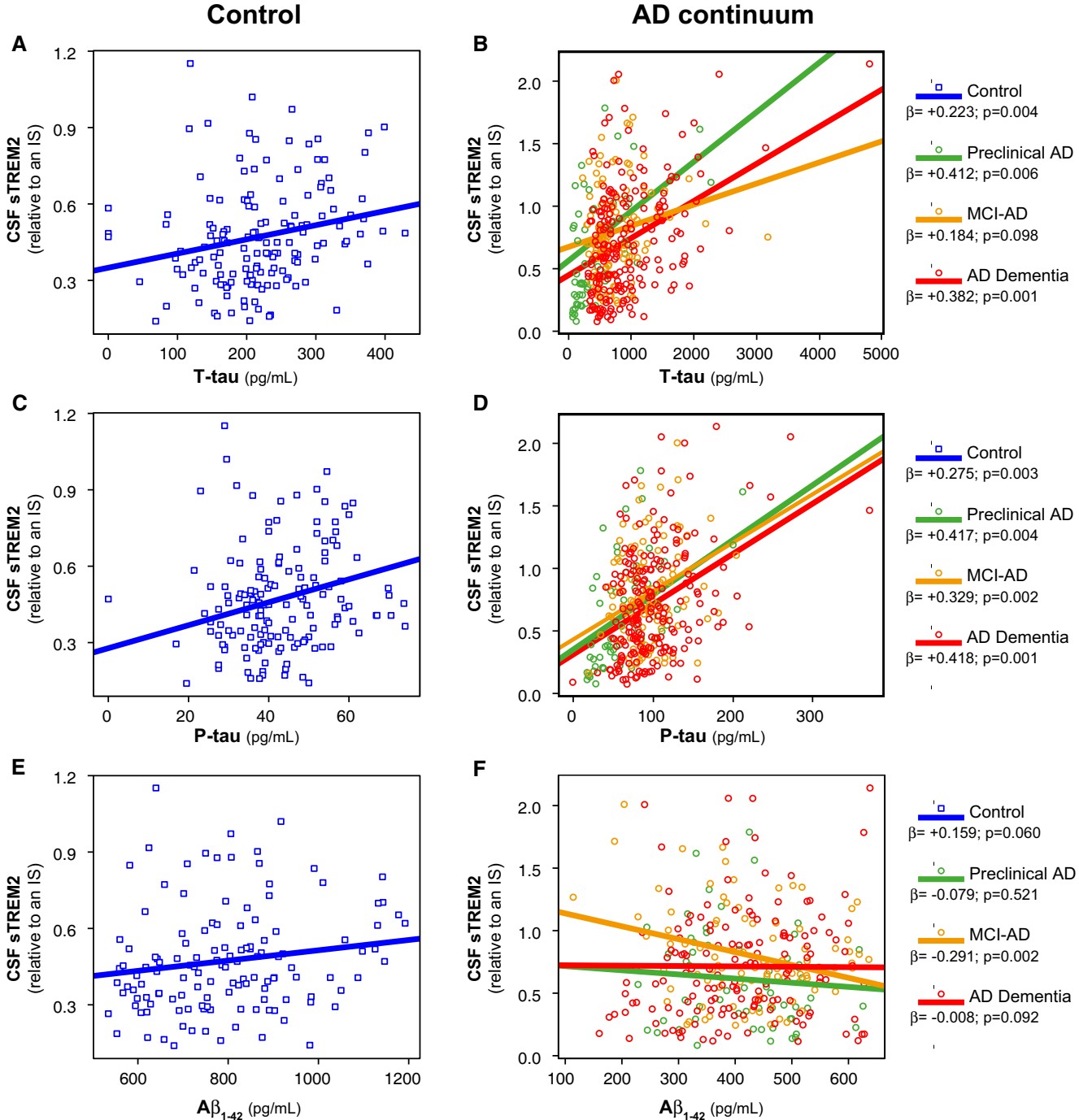

**Figure 3.  Association of CSF sTREM2 with the AD CSF core biomarkers.**

A–F    Scatter plots representing association of CSF sTREM2 with T-tau (A, B), P-tau$_{181P}$ (C, D), and A$\beta_{1-42}$ (E, F) in the control and AD continuum groups. Each point depicts the value of CSF sTREM2 of a subject, and the solid lines indicate the regression line for each of the groups calculated by a linear mixed-effects model adjusted by age and gender (fixed effects) and center (random effects). The standardized regression coefficients ($\beta$) and the $P$-values are also shown. The sample contained some extreme values of T-tau and P-tau$_{181P}$. We did not exclude any value, but we performed a bootstrapping for each association in order to ensure that the associations were not driven by these extreme values.

We compared the MCI-noAD group with the MCI-AD group and the controls, and we observed that the levels of CSF sTREM2 differed within these three groups ($F_{2,349} = 16.7$, $P < 0.0001$). Particularly, MCI-noAD patients showed lower levels of CSF sTREM2 compared to the MCI-AD patients ($P < 0.0001$) but were similar to those of controls (Fig 4B).

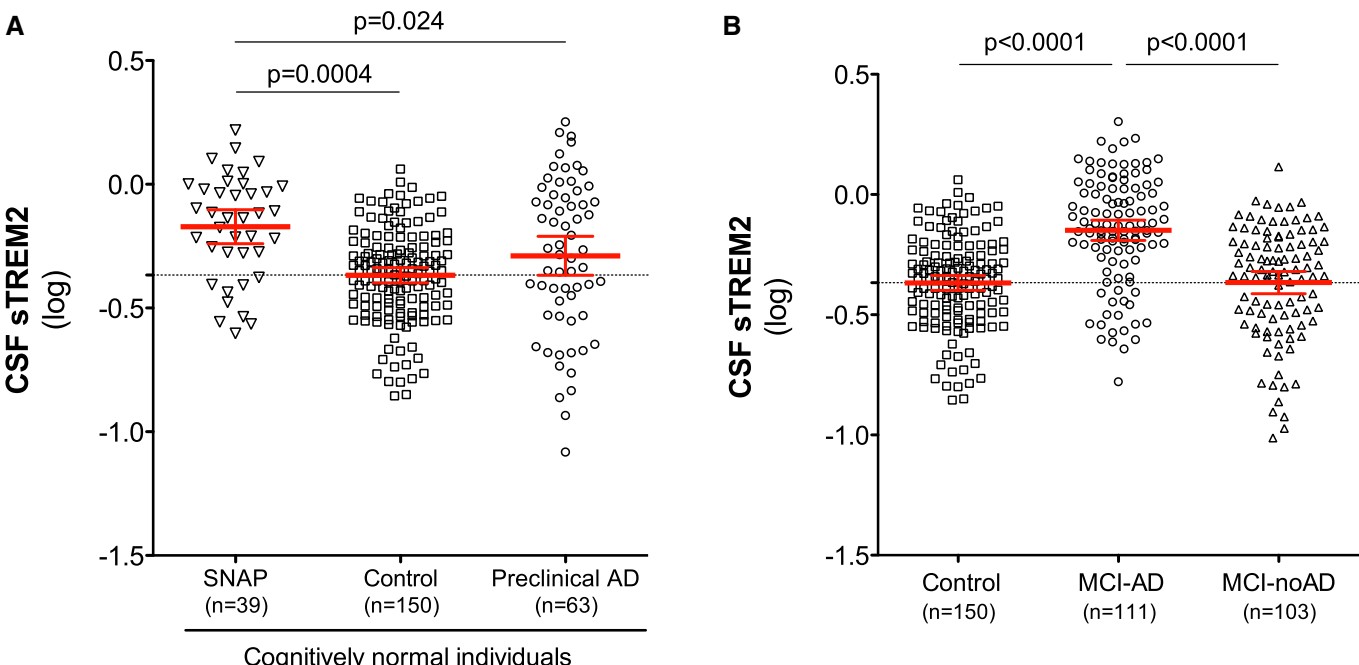

**Figure 4. CSF sTREM2 levels in cognitively normal SNAPs and in MCI-noAD.**

A  Scatter plot showing CSF sTREM2 levels across different groups of cognitively normal individuals: SNAP, suspected non-AD pathophysiology ($n$ = 39); control subjects ($n$ = 150); and preclinical AD ($n$ = 63).

B  Differences in CSF sTREM2 in MCI-AD ($n$ = 111) compared to MCI-noAD ($n$ = 103) and controls ($n$ = 150).

Data information: The red bars represent the mean and the 95% CI. *P*-values were assessed by a linear mixed-effects model adjusting by age and gender (fixed effects) and center (random effect).

When restricting the sample to subjects who were 65 years or older, the results of the regression analysis confirmed that CSF sTREM2 levels were significantly higher in SNAPs than in the control group ($P$ = 0.004) and in MCI-AD than MCI-noAD ($P$ < 0.0001), suggesting that the group differences in CSF TREM2 levels are not due to age differences.

## Discussion

In the present study, we demonstrate that the levels of CSF sTREM2 are elevated in MCI-AD compared to controls and AD dementia subjects. We also found that CSF sTREM2 levels were closely associated with markers of neuronal injury and tau pathology (T-tau and P-tau$_{181P}$), even if there is no evidence of underlying amyloidosis as shown in the SNAP group.

While our manuscript was under consideration, two related studies were published (Heslegrave *et al*, 2016; Piccio *et al*, 2016). Both of them measured CSF sTREM2 in an AD case–control data set and found that sTREM2 is slightly increased in CSF of AD patients. We, in contrast, analyzed CSF sTREM2 in different stages of AD, from preclinical to dementia stages, as a proxy of longitudinal changes. We found that the levels of CSF sTREM2 dynamically change during the AD continuum and peak at the MCI stage of AD. In the dementia stage, CSF sTREM2 levels are still higher than in the control group but that increase does not reach statistical significance after adjusting by age. Thus, our study does not only

expands the previous results in a larger sample, but it also provides the unique finding that the highest levels of CSF sTREM2 occur in early stages of AD progression, coinciding with the appearance of the first symptoms. A similar pattern of change, that is a peak in earlier stages of the disease, has been described for IL-18 and MCP-1 in plasma and in IP-10 in CSF (Galimberti *et al*, 2006a,b; Motta *et al*, 2007; Craig-Schapiro *et al*, 2011; Brosseron *et al*, 2014). Previously, we described that AD dementia patients show a mild but significant decrease in CSF sTREM2 compared to controls (Kleinberger *et al*, 2014), while two recent studies reported an increase of CSF TREM2 in AD dementia (Heslegrave *et al*, 2016; Piccio *et al*, 2016). In the current study, we did not find any significant difference between controls and AD dementia. The likely reason for this discrepancy between studies is that AD dementia severity may have varied between studies.

Our results of increased sTREM2 levels in MCI-AD raise the question if they are a cause or consequence of disease progression. Although this cannot be answered currently, we find a strong association between CSF sTREM2 levels and P-tau$_{181P}$ and T-tau, markers of neuronal and axonal cell injury and neurofibrillary tangles, but only very little if any correlation to A$\beta_{1-42}$. Moreover, CSF sTREM2 levels were also elevated in the SNAP group, which suggests that CSF sTREM2 increases with neuronal injury even in the absence of amyloid pathology. These findings suggest that changes of CSF sTREM2 may indeed be associated with neuronal injury and are consistent with studies that propose that TREM2 recognizes apoptotic neurons and mediates their

phagocytosis (Takahashi et al, 2005; Wang et al, 2015). Given the strong association between CSF levels of sTREM2, P-tau$_{181P}$, and T-tau, one might argue that these markers provide redundant information. In this regard, it is important to highlight the completely different origin of these two proteins. While CSF sTREM2 is physiologically produced and shed by microglia, T-tau and P-tau$_{181P}$ are passively released into CSF by dying neurons. Moreover, sTREM2 production depends on its active transport to the cell surface, where shedding takes place (Kleinberger et al, 2014). This makes CSF sTREM2 a potential candidate of microglial activity. Increased sTREM2 due to microglial activation is consistent with the findings that TREM2 expression (Matarin et al, 2015) and GE-180 uptake (Liu et al, 2015) are increased in elderly mouse models of AD. If the early increase of sTREM2 reflects, a protective or detrimental response remains to be shown. Data obtained from AD mouse models crossed with TREM2 knockouts are controversial by demonstrating either an increase or a decrease of amyloid plaque load (Jay et al, 2015; Wang et al, 2015). However, our finding that CSF sTREM2 increases during normal aging may be in line with a protective response to rather mild neuroinjury. Moreover, dramatically reduced CSF levels of sTREM2 in NHD patients with the T66M mutations (Kleinberger et al, 2014; Piccio et al, 2016) are also in favor of a protective role of sTREM2. In that regard, misfolding of mutant TREM2 prevents its maturation and consequently microglial phagocytosis, which depends on the presence of full-length TREM2 at the plasma membrane (Kleinberger et al, 2014). However, the AD-associated R47H mutations did not show decreased shedding in cellular models (Kleinberger et al, 2014) and was even found to be increased in CSF samples of carriers (Piccio et al, 2016). This clearly demonstrates that disease-associated TREM2 mutations may have different and seemingly opposite cellular mechanisms. However, increased shedding as observed by Piccio et al may also be consistent with reduced cell surface TREM2 and the corresponding functions of membrane-associated TREM2.

The strengths of the current study are the large sample size and the use of both clinical and CSF biomarkers data as a diagnostic criteria. This allowed us to study patients in preclinical stages of AD and to enrich the sample with patients with AD underlying pathology. Many of the inconsistencies observed in biomarkers for neuroinflammation in other studies may be due to the fact that the sample was only clinically characterized and other pathologies besides AD may have been accidentally included. However, our study also has some limitations. First, we found a significant variability in the CSF sTREM2 measurements between centers. Taking into account that the ELISA measurements were performed in the same laboratory, there may be some pre-analytical issues that may influence its measurements that need further investigation. In order to control for the center effect, we include the center as a random covariate in a linear mixed model analysis. Second, we did not screen the included subjects for possible TREM2 mutations. However, it is very unlikely that the possibility of TREM2 mutations in the current patient sample affected our results because TREM2 mutations show a low prevalence in the population and even in AD dementia patients (Guerreiro et al, 2013a; Jonsson et al, 2013). Third, this is a cross-sectional study, thereby limiting any conclusion about

progression. The results should thus be replicated in subjects with longitudinal data to analyze whether CSF sTREM2 levels are associated with disease progression. Such a study may be performed with the DIAN cohort.

# Materials and Methods

### Study design and participants

We conducted a cross-sectional multicenter study in which we studied individuals from five experienced European memory clinics (Appendix Table S3 and S4). The clinical assessment, the lumbar puncture, and the AD CSF core biomarker measurements (Aβ$_{1-42}$, T-tau, and P-tau$_{181P}$) were performed in each of the centers. Taken into account that our aim was to study CSF sTREM2 in different stages of AD, we enriched our sample with AD patients that had both the clinical phenotype of AD and the CSF biomarker profile of AD. In previous studies, the measurement of the AD CSF core biomarkers has clearly demonstrated to increase the accuracy of the diagnosis of AD, to predict whether AD is the underlying pathology responsible of the cognitive symptoms and has shown to be valuable in detecting the earlier stages of the disease (Clark et al, 2003; Shaw et al, 2009; Molinuevo et al, 2014). Likewise, the normality of the AD CSF core biomarkers is useful to exclude underlying amyloidogenesis and/or neurodegeneration and hence recruit a more specific control group. Therefore, the diagnostic criteria of the present study were based both on the clinical diagnosis and on the AD CSF core biomarker profile, following NIA-AA recommendations (Albert et al, 2011; McKhann et al, 2011; Sperling et al, 2011). The cutoff values to define abnormal CSF values for each biomarker were defined for each center and are displayed in the Appendix Table S1.

The control group consisted of asymptomatic and cognitively normal elderly subjects who were recruited by different strategies depending on the centers and described in detail in the Appendix Supplementary Methods. However, all controls fulfilled at least the following criteria: (i) no cognitive complaints (hence, subjective cognitive decline, SCD, were not included); (ii) cognitive deterioration was ruled out after evaluation by a neurologist and/or by means of neuropsychological screening; (iii) no evidence of stroke, neuroinflammatory, or neurodegenerative diseases according to the evaluation by a neurologist; (iv) normal levels of all three AD CSF core biomarkers. The control group also contained 22 patients with psychiatric or other neurological diseases unrelated to the CNS (see Appendix Supplementary Methods for detailed description of the control group).

Following the NIA-AA criteria (Sperling et al, 2011), we defined preclinical AD in terms of normal cognitive test performance as cognitively normal subjects (asymptomatic or with SCD) and decreased CSF Aβ$_{1-42}$. The classification SCD was assigned to subjects presenting with memory complains but without an objective cognitive impairment. We did not use T-tau or P-tau$_{181P}$ as criteria in this group in order to also include the earliest preclinical AD stage characterized by decreased Aβ$_{1-42}$ but not yet increased T-tau or P-tau$_{181P}$.

Patients with mild cognitive impairment due to AD (MCI-AD) were classified according to the NIA-AA criteria (Albert et al,

2011). In brief, these subjects were clinically diagnosed as MCI (following standard criteria) (Winblad *et al*, 2004) or with CDR = 0.5 or CDR-SOB = 0.5–4, in combination with decreased $A\beta_{1-42}$ and increased T-tau and/or P-tau$_{181P}$. Thus, the criteria for "MCI due to AD" of high likelihood were fulfilled (Albert *et al*, 2011). Likewise, AD dementia was defined following NIA-AA criteria (McKhann *et al*, 2011) for probable AD, with the requirment of decreased $A\beta_{1-42}$ and increased T-tau and/or P-tau$_{181P}$. Therefore, the criteria for probable AD dementia with high evidence of AD pathophysiological process were fulfilled (McKhann *et al*, 2011). All clinical diagnoses were made by neurologists with expertise on neurodegenerative diseases. We did not include age as an inclusion criterion since we did not know *a priori* if CSF sTREM2 was influenced by age. However, all analyses were adjusted by age.

In addition to the subjects included in the main analysis, we also received and measured CSF samples of subjects who did not fulfill the diagnostic criteria for the control group or preclinical AD, MCI-AD, or AD dementia groups. These samples comprise cognitively normal subjects with increased T-tau and/or P-tau$_{181P}$ (cognitively normal SNAPs) (Jack *et al*, 2012) and clinically diagnosed MCI subjects who did not fulfill the NIA-AA criteria for "MCI due to AD" with high likelihood, namely they did not accomplish the requisite of decreased $A\beta_{1-42}$ in combination with increased T-tau and/or P-tau$_{181P}$, and referred in the text as MCI-noAD. Appendix Tables S3 and S4 depict a complete list of the groups along with summary statistics of their demographic and clinical data.

In total, we studied CSF of 706 individuals, but we excluded 40 of them from the analysis due to different reasons: (i) missing data in 30 subjects, since we only included those subjects for which the following data were available: age, gender, clinical diagnosis, and CSF levels of $A\beta_{1-42}$, T-tau, and P-tau$_{181P}$; the excluded subjects did not have different CSF sTREM2 levels than the rest of the participants; (ii) seven subjects had a CSF sTREM2 measurement in the ELISA with an intraplate coefficient of variation (CV) > 15%; (iii) 1 control and 2 AD dementia subjects were considered as outliers defined as CSF sTREM2 levels > 3 standard deviations (SD) below or above the group mean CSF sTREM2 level. Among the 666 samples analyzed, 374 of them fulfilled the NIA-AA criteria for one of AD continuum stages (including 63 preclinical AD, 111 MCI-AD, and 200 AD dementia) and 150 were considered as controls. *APOE* was genotyped in these patients by standard methods in each participating center, and it was available in 74% of the subjects. The sample studied also contained the following diagnosis: cognitively normal SNAPs (*n* = 39) and MCI-noAD (*n* = 103).

## CSF collection and biochemical procedures

CSF samples were obtained by lumbar puncture following standard procedures, collected in polypropylene tubes, and immediately frozen at −80°C until use. All centers participating use a similar standardized operating procedure (SOP) for pre-analytical sample handling and follow international consensus recommendations (Blennow *et al*, 2010; Vanderstichele *et al*, 2012).

The values of $A\beta_{1-42}$, T-tau, and P-tau$_{181P}$ were provided by each participating center; all of them have experience in CSF biomarker determination and have participated in the Alzheimer's Association external quality control program (Mattsson *et al*,

2011) and/or the Alzheimer's Biomarkers Standardization Initiative (ABSI) for CSF biomarkers (Molinuevo *et al*, 2014). In all centers, these biomarkers were measured by the commercially available INNOTEST ELISA kits for $A\beta_{1-42}$ (INNOTEST β-amyloid$_{1-42}$; Fujirebio Europe), T-tau (INNOTEST hTAU Ag; Fujirebio Europe), and P-tau$_{181P}$ (INNOTEST Phospho-Tau$_{181P}$; Fujirebio Europe), except for the Bonn group that measured $A\beta_{1-42}$ and T-tau with the MesoScale Discovery platform (MSD, Gaithersburg, MD, USA). As already reported, $A\beta_{1-42}$ measurements were higher using the MSD platform (Mattsson *et al*, 2011). Therefore, we excluded them in the association analysis between CSF sTREM2 and $A\beta_{1-42}$.

## Soluble TREM2 (sTREM2) measurement

CSF sTREM2 was measured by an ELISA previously established by our group using the MSD Platform (Kleinberger *et al*, 2014). The characteristics of the ELISA are described in detail in the Appendix Supplementary Methods. All the samples were measured in duplicate, and the operator was blinded to the clinical diagnosis. The mean intraplate CV was 2.9% and the interplate CV 12.9%. Duplicate measures with an intraplate CV > 15% were discarded. A dedicated CSF sample (internal standard, IS) was loaded in all plates, and in order to account for the interplate variability, all the measurements were expressed in relation to the IS of each plate. The absolute values (ng/ml) are reported in the Appendix Table S3.

## Statistical analysis

Differences in the demographic data and the CSF core biomarkers between diagnostic groups were assessed by Pearson chi-square test for categorical variables, and one-way ANOVA for continuous variables followed by Bonferroni *post hoc* tests. The association between CSF sTREM2 and age was studied with Pearson product-moment correlation test.

To investigate the differences in CSF sTREM2 between the diagnostic groups or other group categories (gender, *APOE* ε4 status), we first log$_{10}$-transformed the outcome variable (CSF sTREM2) to approach the assumptions of Gaussian normal distribution. In order to assess whether there are differences in CSF sTREM2 levels between control, preclinical AD, MCI-AD, and AD dementia groups, we performed a linear mixed-effects regression analysis with CSF sTREM2 as dependent variable and diagnostic group, gender, and age as independent fixed variables. To control for the intercenter variation, center was introduced in the mixed model as a random intercept effect. *Post hoc* tests were used for pair-wise comparisons of CSF TREM2 levels between the diagnostic groups, using Bonferroni correction. In order to confirm the robustness of the results, we also calculated the CSF sTREM2 means and 95% confidence intervals (CI) in each diagnostic group by bootstrapping with 1,000 resampling iterations. The same approach was implemented for assessing differences in sTREM2 between the SNAP and the MCI-noAD groups. The association of CSF sTREM2 and the AD CSF core biomarkers were also studied with a linear mixed-effects model with age and gender as a fixed effects and the center as a random effect. The standardized regression coefficients ($\beta$) are reported. In order to rule out that the associations were

**The paper explained**

**Problem**

TREM2 is a transmembrane protein selectively expressed by microglia in the central nervous system, and its ectodomain is cleaved and released into the extracellular space as a soluble variant (sTREM2), which can be detected in CSF. Certain *TREM2* mutations, which are associated with neurodegenerative disorders (including AD), reduce shedding of TREM2 and are therefore believed to cause a loss-of-function. This may suggest a change of microglial TREM2 expression and processing in response to neuronal injury. However, during the progression of AD, microglia may present with different phenotypes, which can have a beneficial or a detrimental role in the disease. In the present study, we therefore investigated how CSF sTREM2 levels change during the course of AD and if these changes correlate with markers for neuronal injury.

**Results**

We report that levels of CSF sTREM2 change during the course of AD. Specifically, they reach the highest levels in the MCI stage and this increase is attenuated in the dementia stage. Moreover, increased CSF sTREM2 levels are associated with higher levels of T-tau and P-tau$_{181P}$, markers of neuronal cell injury, and neurofibrillary tangles.

**Impact**

Our findings suggest that levels of CSF sTREM2 reflect a microglial response to early cell death. These findings reinforce the idea that the inflammatory and microglial response change during the progression of AD.

driven by extreme values, the *P*-values were again calculated by bootstrapping.

Statistical analysis was performed in SPSS IBM, version 20.0, statistical software, and the free statistical software R (http://www.r-project.org/). All tests were two-tailed, with a significant level of $\alpha = 0.05$. Figures were built using GraphPad Prism or the free statistical software R.

**Patient consent**

All participants or their relatives gave their written consent. The ethics committee at each center approved the study and was in accordance with the Declaration of Helsinki.

**Expanded View** for this article is available online.

## Acknowledgements

We are grateful to Nadine Pettkus and Brigitte Nuscher for technical assistance. This work was supported by the European Research Council Under the European Union's Seventh Framework Program (FP7/2007–2013)/ERC Grant Agreement No. 321366-Amyloid, the Deutsche Forschungsgemeinschaft (German Research Foundation) within the framework of the Munich Cluster for Systems Neurology (EXC 1010 SyNergy), Cure Alzheimer's Fund, and MetLife Foundation Award (to Christian Haass). Kaj Blennow was funded by the Swedish Research Council and the Torsten Söderberg Foundation at the Royal Swedish Academy of Sciences. Henrik Zetterberg was funded by the Swedish Research Council, the Knut and Alice Wallenberg Foundation, and Frimurarestiftelsen. Sebastiaan Engelborghs was in part supported by the University Research Fund of the University of Antwerp; the Institute Born-Bunge; the Foundation for Alzheimer Research (SAO-FRA); Neurosearch Antwerp; the Research Foundation—Flanders (FWO); the Agency for Innovation by Science and Technology (IWT); the Interuniversity Attraction Poles (IAP) Program of the Belgian Science Policy Office; the Flemish Government Methusalem Excellence Program, Belgium; the Flanders Impulse Program on Networks for Dementia Research (VIND). This work is part of the BIOMARKAPD project within the EU Joint Program for Neurodegenerative Disease Research (JPND). This work has received support from the EU/EFPIA Innovative Medicines Initiative Joint Undertaking (EMIF grant no. 115372). The genetics at the VIB, Antwerp site was in part funded by the Belgian Science Policy Office Interuniversity Attraction Poles Program, the Flanders Government Initiated Impulse Program on Networks for Dementia Research (VIND), the Flemish government Initiated Methusalem Excellence Program, and the University of Antwerp Research Fund, Belgium.

## Author contributions

MSC and GK performed the experiments. MSC, GK, MAC, MB, AR, JDG, KS, CVB, AC, ME, and CH analyzed and interpreted the data. DA, JF, AL, RB, JDG, RSV, AA, LR, JLM, FB, AT, MTH, HS, SE, KS, CVB, HZ, BN, and KB contributed with patient samples and data. MSC, GK, MAC, ME, and CH designed the study, interpreted the data, and wrote the manuscript. All authors critically reviewed and approved the final manuscript.

## Conflict of interest

Sebastiaan Engelborghs is consultant for and received research funding from Janssen, ADx Neurosciences, Innogenetics/Fujirebio Europe, Lundbeck, Pfizer, Novartis, UCB, Roche Diagnostics, Nutricia/Danone. The remaining authors declare that they have no conflict of interest.

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
