## [Review Process File · EMBO Molecular Medicine]

sTREM2 cerebrospinal fluid levels are a potential biomarker for microglia activity in early-stage Alzheimer's disease and associate with neuronal injury markers

Marc Suárez-Calvet, Gernot Kleinberger, Miguel Ángel Caballero, Matthias Brendel, Axel Rominger, Daniel Alcolea, Juan Fortea, Alberto Lleó, Rafael Blesa, Juan Domingo Gispert, Raquel Sánchez-Valle, Anna Antonell, Lorena Rami, José L. Molinuevo, Frederic Brosseron, Andreas Träschütz, Michael T. Heneka, Hanne Struyfs, Sebastiaan Engelborghs, Kristel Slegers, Christine Van Broeckhoven, Henrik Zetterberg, Bengt Nellgård, Kaj Blennow, Alexander Crispin, Michael Ewers, Christian Haass

Corresponding authors: Christian Haass and Michael Ewers, Ludwig-Maximilians-University

Review timeline:	Submission date:	03 December 2015
	Editorial Decision:	21 January 2016
	Revision received:	27 January 2016
	Accepted:	28 January 2016

Transaction Report:

Editor: Céline Carret

1st Editorial Decision

21 January 2016

Thank you for the submission of your manuscript to EMBO Molecular Medicine. We have now received the enclosed reports from the three referees who were asked to assess it. As you will see the reviewers are globally supportive and only request a minor revision to discuss a few points and rewrite maybe the discussion section. I am pleased to inform you that we will be able to accept your manuscript pending editorial final amendments.

1) please address the reviewers concerns and provide a letter INCLUDING the reviewer's reports and your detailed responses to their comments (as Word file).

Please submit your revised manuscript as soon as possible but within 3-months.

I look forward to receiving the revised article.

***** Reviewer's comments *****

Referee #1 (Remarks):

The authors present a large cross-sectional study with samples collected at multiple sites to examine the CSF levels of soluble TREM2 (sTREM2) from controls, pre-AD, MCI-AD, AD with dementia, MCI-non-AD (mild cognitive impairment without AD CSF biomarker changes), and SNAP (suspected non-AD pathology; cognitively normal individuals with tau biomarker changes in CSF). Soluble TREM2 is released upon cleavage of TREM2 on microglia and may reflect microglial activation. CSF sTREM2 increased significantly with age but was not influenced by gender or ApoE E4 status within each group. CSF sTREM2 levels were significantly elevated in MCI-AD vs. controls and AD with dementia. CSF sTREM2 levels were higher in MCI-AD vs. pre-AD but statistical analysis revealed a strong trend. No such increase was observed in MCI-non-AD CSF. Unlike the previous report from the authors in which AD patients had lower CSF sTREM2, there was no significant difference in sTREM2 CSF levels between AD and controls in this study. In fact, CSF sTREM2 levels were non-significantly higher in AD CSF vs. controls (adjusted for gender, age and site). This result may be due to the high degree of variability in the CSF sTREM2 levels in samples collected at different sites. CSF sTREM2 levels positively correlated with CSF total tau and p-tau in MCI-AD, suggesting that sTREM2 levels may reflect an early microglial response to neuronal injury. Interestingly, CSF sTREM2 levels were significantly increased in SNAP CSF, again suggesting a link to neurodegenerative processes.

The paper provides an excellent, first-ever evaluation of CSF soluble TREM2 in the AD continuum and in SNAP. The data are in agreement with other studies that suggest that early microglial activation plays a role in AD pathogenesis and cognitive impairment. This study offers an important contribution to the field of neurodegeneration in general, and AD, in particular, and will undoubtedly lead to further research into mechanisms underlying the connection between early microglial activation and neurodegeneration.

Comments:

1. Is the TREM2 genotype available for the subjects in this study? Is it possible that the difference in the AD results (from the previous study) might be due to fewer patients with a TREM2 variant, which would be predicted to have reduced TREM2 expression? As the authors noted in the Introduction, TREM2 variants are fairly common in AD. Do the authors predict that the CSF sTREM2 levels would be different in an MCI-AD patient with a TREM2 variant vs. an MCI-AD patient without a TREM2 variant? Please discuss.
2. The authors suggest that increased CSF sTREM2 levels reflect a response to neurodegeneration. Is it possible that microglial activation precedes (and drives) neurodegeneration?
3. Optional: How do these results line up with the differences in the TREM2 KO mouse publications? For example, could the disease stage in the mice have influenced the results? Or, is it possible that we need to wait for the Tau Tg/TREM2 KO to see an effect?
4. Please check the references for accuracy of spelling (e.g. authors' names and initials).

Referee #2 (Remarks):

The authors performed an ambitious, multicenter study examining whether CSF sTREM2 levels were different in individuals representing the spectrum of Alzheimer's disease, from controls to preclinical AD to mild cognitive impairment (MCI) to dementia. This study faced some significant challenges but accounted for these well in their analyses. Part of the difficulty of a multicenter study on CSF biomarkers is that each center has different criteria for controls, CSF collections protocols, CSF biomarker cut-offs, etc. Although there were important differences between centers, this group did a good job of documenting the relevant center-specific information and controlled for a center effect by adding it into their models.

The authors found that CSF sTREM2 levels increased with age and this effect was found in both controls and individuals on the AD spectrum. Levels of CSF sTREM2 were highest in the MCI-AD and AD dementia groups. Interestingly, levels of CSF sTREM2 were lower in the AD dementia

group than in the MCI-AD group, suggesting that sTREM2 peaks with the onset of cognitive symptoms and then later declines. If this is true, it would be important because high sTREM2 levels might identify individuals with incipient dementia. The study had a potential weakness because their control group was a decade younger than the other groups. Although they included age in their models, they further evaluated this potential confound by repeating their analysis in age-matched groups and convincingly found that the MCI-AD group still had higher sTREM2 levels than controls. The authors also found correlations between CSF sTREM2 and CSF Tau, pTau and A 42. As might be expected, CSF sTREM2 levels correlated better with CSF Tau and pTau levels, again suggesting that elevated sTREM2 levels occur later in the course of AD pathology.

The authors also evaluated the CSF sTREM2 levels of individuals with evidence of neurological dysfunction not related to Alzheimer's disease. These analyses included either cognitively normal individuals with normal CSF A 42 but high CSF tau or ptau (classified as Suspected Non-Alzheimer's Pathology [SNAP]) or with MCI but CSF biomarker not consistent with AD (MCI-no AD). Interestingly, individuals classified as SNAP had elevated sTREM2. Individuals with MCI-AD had higher levels of sTREM2 than individuals with MCI-no AD. The authors should include a control group in this MCI-AD and MCI-no AD analysis (Fig. 4B), because it is not clear whether levels of sTREM2 in MCI-no AD are normal.

The authors do need to significantly change their discussion (and some parts of their introduction) because two papers have come out in the past month that examine CSF sTREM2 in AD and have highly relevant and supportive results to this work. The first is by Piccio et al. in *Acta Neuropathologica* and the second is by Heslegrave in *Molecular Neurodegeneration*.

Piccio L, et al.. Cerebrospinal fluid soluble TREM2 is higher in Alzheimer disease and associated with mutation status. *Acta Neuropathol.* 2016 Jan 11. [Epub ahead of print]

Heslegrave A, Heywood W, Paterson R, Magdalinou N, Svensson J, Johansson P, +hrfelt A, Blennow K, Hardy J, Schott J, Mills K, Zetterberg H. Increased cerebrospinal fluid soluble TREM2 concentration in Alzheimer's disease. *Mol Neurodegener.* 2016 Jan 12;11(1):3. doi: 10.1186/s13024-016-0071-x

Despite the overlap between this manuscript and the Piccio/Heslegrave papers, this topic is highly impactful and replication advances the field. Further, this study represents a larger sample than either the Piccio/Heslegrave papers and has the unique and important finding that sTREM2 levels may change along the AD spectrum, peaking in MCI-AD and then declining.

Referee #3 (Remarks):

This is a very well done study looking at Soluble Trem2 levels in CSF among individuals with AD, MCI, SNAP and preclinical AD. CSF was obtained from multiple sites and the assay shows excellent performance. While, I find the data compelling with the exception of one small quirk (noted explicitly below) I do find the discussion both a little long and a little overreaching in terms of possible implications of this data.

Indeed, how sTREM2 relates to microglial activation states is not clearly established. It is shed but what regulates that shedding Especially in isolation it is unlikely that sTREM2 will be that informative as microglial cells produce hundreds of secreted proteins. Data describing how sTREM2 levels are altered by various immune stimuli might make this an article of broader impact. Additionally looking at other innate immune markers might be very important to put this data in context.

The data from Saint PAs cohort is somewhat problematic as it is almost the opposite of the other data. Lower levels in the AD continuum subjects. If there is this kind of site to site variability this may be a very hard finding to reproduce. Some discussion of this is warranted and perhaps some investigation into possible confounds from various sites

Please find attached the revised version of the manuscript EMM-2015-06123. We have now carefully addressed all points raised by the reviewers and specifically re-written the discussion with a special emphasis on the new data presented in the communications published while our paper was under consideration. In detail we addressed the points of the reviewers as follows:

Reviewer 1:

1) Is the TREM2 genotype available for the subjects in this study?

We have not screened the individuals included in the study for *TREM2* mutations. However, the prevalence of *TREM2* mutations is rare (minor allele frequency <1%) (Guerreiro et al 2013; Jonsson et al. 2013). Therefore, it is extremely unlikely that our sample contains a significant number of *TREM2* mutant carriers that may affect our results. Moreover, the main analyses have been confirmed by Bootstrapping in order to control for the effect of any potential outlier. However, we agree with the reviewer that this is a limitation of the study and, therefore, we have mentioned it in the discussion section. The reviewer stated that we mentioned in the introduction that “*TREM2* variants are fairly common in AD”. We like to clarify that we noted in the introduction that *TREM2* mutations are associated with increased risk of AD. However, this does not imply that the mutations are prevalent in AD dementia.

2) The authors suggest that increased CSF sTREM2 levels reflect a response to neurodegeneration. Is it possible that microglial activation precedes (and drives) neurodegeneration? Optional: How do these results line up with the differences in the TREM2 KO mouse publications?

Both points are now specifically discussed on page 10 and 11 of our revised manuscript. Please note that we have re-written and streamlined the entire discussion as requested by reviewers 2 and 3.

3) Please check the references for accuracy

All references were checked for accuracy.

Reviewer 2:

1) Individuals with MCI-AD had higher levels of sTREM2 than individuals with MCI-no AD. The authors should include a control group in this MCI-AD and MCI-no AD analysis (Fig. 4B), because it is not clear whether levels of sTREM2 in MCI-no AD are normal.

We have now added the control group to the new Fig. 4B as requested.

2) The authors do need to significantly change their discussion (and some parts of their introduction) because two papers have come out in the past month that examine CSF sTREM2 in AD and have highly relevant and supportive results to this work.

We have now extensively re-written the discussion also in accord to reviewer's 3 comments (see below). This includes a detailed comparison of our concept to investigate the AD continuum versus the rather simple comparison of AD (a mixture of all stages) with controls. Moreover, reviewer 3 found our discussion a bit long and overreaching. In accordance with that we reduced the discussion by 50% and streamlined it to the most important points (please refer to the completely new discussion section).

Reviewer 3:

1) I do find the discussion both a little long and a little overreaching in terms of possible implications of this data.

Please refer to the similar point raised by reviewer 2 (and also reviewer 1) and the completely new discussion section.

2) The data from Saint PAs cohort is somewhat problematic as it is almost the opposite of the other data.

We agree with the reviewer that there is significant variability between centers and it needs to be addressed in the future which pre-analytical issues may influence CSF sTREM2 measurement, as it has been done with other CSF biomarkers. This is a limitation of the study and we have clearly highlighted that in the discussion section. In order to control for the center effect, we used a linear mixed effects model with center as a random effect. Noteworthy, the highest levels of CSF sTREM2 in all centers occur in the MCI-AD group (also in Sant Pau center) as depicted in Appendix Table S4, with the only exception of the Bonn center in which the highest levels occurs in the preclinical stage.

Taken together we believe that we have carefully addressed all points raised by the reviewers. We are now looking forward to the publication of our findings in EMBO Mol Med. Again, many thanks for considering our manuscript.

Corresponding Author Name: CHRISTIAAN HAASS / MICHAEL EWERS
 Journal Submitted to: EMBO MOLECULAR MEDICINE
 Manuscript Number: